# Metformin in Differentiated Thyroid Cancer: Molecular Pathways and Its Clinical Implications

**DOI:** 10.3390/biom12040574

**Published:** 2022-04-14

**Authors:** Manuel García-Sáenz, Miry Lobaton-Ginsberg, Aldo Ferreira-Hermosillo

**Affiliations:** 1Servicio de Endocrinología, Hospital de Especialidades del Centro Médico Nacional Siglo XXI, Instituto Mexicano del Seguro Social, Mexico City 06720, Mexico; manuel.gsm@hotmail.com; 2Unidad de Investigación Médica en Enfermedades Endócrinas, Centro Médico Nacional Siglo XXI, Instituto Mexicano del Seguro Social, Mexico City 06720, Mexico; lobatonmiry@gmail.com

**Keywords:** metformin, thyroid cancer, pharmacological mechanisms of action, clinical pathways

## Abstract

Metformin is a synthetic biguanide that improves insulin sensitivity and reduces hepatic gluconeogenesis. Aside being the first-line therapy for Type 2 Diabetes (T2D), many pleiotropic effects have been discovered in recent years, such as its capacity to reduce cancer risk and tumorigenesis. Although widely studied, the effect of metformin on thyroid cancer remains controversial. Potential mechanisms for its growth inhibitory effects have been elucidated in various preclinical studies that involved pathways related to adenosine mono-phosphate-activated protein kinase (AMPK), mammalian target of rapamycin (mTOR), mitochondrial glycerophosphate dehydrogenase (mGPDH), and the nuclear factor κB (NF-κB). Hyperinsulinemia increases cell glucose uptake and oxidative stress, and promotes thyroid cell growth, leading to hyperproliferation, carcinogenesis, and the development of malignant tumors. Furthermore, it has also been related to thyroid nodules size in nodular disease, as well as tumoral size in patients with thyroid cancer. Several clinical studies concluded that metformin might have an important role as an adjuvant therapy to reduce the growth of benign and malignant thyroid neoplasms. This suggests that metformin might be useful for patients with differentiated or poorly differentiated thyroid cancer and metabolic diseases such as insulin resistance or diabetes.

## 1. Introduction

Metformin is a synthetic biguanide derived from the French lilac (*Galega officinalis*) which has guanidine as an active component. It improves insulin sensitivity and reduces hepatic gluconeogenesis [1]. Metformin is not metabolized by cytochromes P450 in the liver, so drug transporters have the major role of its pharmacokinetics. Due to its hydrophilicity, metformin cannot simply diffuse through cell membranes, so is transported into the cell via uptake transporters as the organic cation transporter 1 (OCT1), which is primarily expressed in the hepatocytes [2]. It is excreted via active tubular secretion in the kidneys, with a half-life of ~5 h.

The uptake from circulation to the renal epithelium is facilitated by organic cation transporter 2 (OCT2). Meanwhile, its excretion to the lumen is mediated through human multidrug and toxin extrusion (MATE)-1 and MATE2-K, which are expressed in the apical membrane of the renal proximal tubule cells [3]. Its maximum recommended dose for treatment of type 2 diabetes (T2D) is 2.5 g per day (35 mg/kg body weight). Based on experiments in animal models and positron emission tomography (PET) in humans, it is estimated that the metformin concentration in the human liver is about 50–100 μM.

Aside being the first-line therapy for most of the patients with T2D, many pleiotropic effects have been discovered in recent years, such as its capacity to reduce cancer risk and tumorigenesis [2]. The interest on metformin for cancer prevention and treatment is based on clinical studies that showed that it is associated with significantly lower cancer incidence in patients with diabetes [4,5,6,7,8]. It has been proposed that metformin inhibits cellular proliferation, has anti-inflammatory and anti-angiogenic effects, which inhibits stem-cells, and has immunomodulatory effects on tumoral cells, which inhibits DNA damage. Many clinical trials have proved its positive effects on lung, liver, pancreas, endometrium, colorectal, breast, prostate, and bladder cancer [1,2]. Nevertheless, information regarding its effect on endocrine cancers is limited [9].

The incidence of endocrine tumors has increased worldwide. Thyroid cancer is the most common endocrine malignancy and, recently, its prevalence and incidence have risen; meanwhile, its mortality has remained low. That increase could be associated with insulin resistance (IR) [9,10]. According to worldwide Globocan 2020 statistics, thyroid cancer reached ninth place with 586,202 new cases (incidence of 6.6/100,000) and represents the 24th place in the mortality rate (0.43/100,000). This cancer was more common in Asia, in the female population, and in those aged among 35–64 years [11,12].

Thyroid cancer is classified in two categories based on its origin: epithelial-derived and neuroendocrine C-cell-derived medullary thyroid cancer (MTC). Furthermore, epithelial-derived cancer is subclassified into differentiated cancer (DTC) that includes: papillary (PTC), follicular (FTC), and Hürthle cell thyroid cancer (HCTC); poorly differentiated thyroid cancer (PDTC); and anaplastic thyroid cancer (ATC) [9]. PTC is the most common histologic subtype, accounting for 90% of new cases and has the best prognosis. Some of the risk factors to develop thyroid cancer are exposure to ionizing radiation in the head and neck, family history, female sex, advanced age, iodine deficiency or excess, and tobacco and alcohol use. Recently, obesity, insulin resistance (IR), hyperinsulinemia, diabetes mellitus, and other metabolic disturbances have been associated with a higher incidence of thyroid carcinoma [13,14].

## 2. Molecular Insights of Metformin in Thyroid Cancer

Although widely studied, the effect of metformin on thyroid cancer remains controversial. Potential mechanisms for its growth inhibitory effects have been elucidated in various preclinical studies [9]. Experimental studies have shown that metformin can influence the growth and progression of thyroid cancer cells. Those studies demonstrated that metformin has different molecular targets that are involved in the control of several metabolic and inflammatory pathways, which can be divided into a) direct effects, related with its mechanism of action (as those related to adenosine mono-phosphate-activated protein kinase (AMPK), mammalian target of rapamycin (mTOR), mitochondrial glycerophosphate dehydrogenase (mGPDH), and the nuclear factor κB (NF-κB)) that induce energetic stress by mimicking a state of caloric deprivation; or b) indirect effects, which result from the interaction with other metabolic and hormonal systems (such as its effects on the thyroid-stimulating hormone (TSH)) [15,16].

### 2.1. Proposed Direct Antitumor Mechanisms Related to Metformin in DTC

In thyroid cancer cell lines, experimental studies have demonstrated that metformin inhibits cell proliferation and promotes apoptosis through AMPK activation, mTOR signaling inhibition, and cell cycle arrest [17,18,19,20,21]. AMPK is an energy sensor and a major regulator of cellular energy homeostasis. The activation of AMPK triggers its downstream target, the tuberous sclerosis complex 2 (TSC2), which inhibits the mTOR signaling pathway, and hampers the activation of ribosomal protein S6 kinase beta-1 (p70S6K/pS6), necessary for cancer cell growth [9]. Moreover, metformin-mediated AMPK activation phosphorylates an inhibitory serine residue in the insulin receptor substrate-1 (IRS-1). IRS-1 is a downstream mediator of insulin-like growth factor 1 receptor (IGF-1R), which transmits signals to the phosphatidylinositol 3-kinase/protein kinase pathway (PI3K/AKT) [22]. More recently, metformin has been associated with the tumor necrosis factor receptor 1 (TNFR1) and G1/S checkpoint regulation signaling pathways [23], involved in the reduction in cancer cell growth. Furthermore, metformin could also regulate cancer cell biology in an AMPK-independent way through the inhibition of the unfolded protein response (UPR) that consequently causes apoptosis, prevents angiogenesis, and induces toxicity on cancer stem cells [24].

Thyroid hormones and TSH can influence the biological effects of growth hormone and IGF-1 on its target tissues. IGF-1 supports normal function, volume, and hormone synthesis in the thyroid gland, but it also participates in pathological conditions, including thyroid enlargement and tumorigenesis, as observed in acromegaly. It has been observed that IGF-1 stimulates thyroid cell proliferation and differentiation [25]. Furthermore, signaling through IGF-1 and insulin receptors is essential for thyroid epithelial survival and vitality due to interactions of their downstream pathways with those of the TSH receptor [26]. Among the pathways activated by IGF-1/insulin signaling in thyroid cells are extracellular signal-regulated kinase 1/2 (Erk1/2) and Akt. Examination of IGF-1/insulin receptor docking proteins has revealed a role for insulin receptor substrate 2 (IRS-2) in mediating the proliferative actions of IGF-1, both in vitro and in vivo [27]. Furthermore, IGF-1 and cyclic adenosine monophosphate (cAMP) differentially activate the PI3 kinase pathway, leading to G1 cyclin-cyclin-dependent kinase activation and DNA synthesis. IGF-1 also induces several anti-apoptotic proteins in thyroid cells, including Fas-associated death domain-like interleukin-1-converting enzyme-inhibitory protein, through activation of the NF-κB pathway [25].

In the human papillary thyroid carcinoma cells line (TPC1), metformin reduced p-ERK, a member of the mitogen-activated protein kinase (MAPK) family associated with proliferation and cell survival. Metformin induced p-AKT in TPC1 and human follicular thyroid cancer cells (FTC236) via PI3K/AKT signaling activated by rearrangement in the transformation/papillary thyroid carcinomas gene (RET/PTEC) and phosphatase and tensin homolog (PTEN) mutation, respectively. Metformin also increased the expression of binding of immunoglobulin protein (BIP), CCAAT/enhancer-binding protein homologous protein (CHOP), and caspase-12 markers of endoplasmic reticulum (ER) stress, another apoptotic mechanism [28].

On other hand, metformin inhibits mitochondrial respiration through inhibition of mitochondrial complex 1 and mGPDH, an enzyme present on the outer surface of the inner mitochondrial membrane that is part of the glycerol-3-phosphate shuttle. The inhibition of mGPDH activity and expression is associated with a reduction in oxidative phosphorylation, which has a negative impact on thyroid cancer cell growth [29].

Other mechanisms involve the inflammatory state, especially due to an increase in cytokines such as interleukin 8 (CXCL8) and tumor necrosis factor alpha (TNF-α). CXCL8 is a mediator of NF-κB signaling, mainly related to the growth and progression of thyroid cancer [30,31]. CXCL8 directly stimulates the proliferation of thyroid tumor cells via autocrine and paracrine mechanisms mediated by its chemokine receptors CXCR1 and CXCR2, which are expressed only in tumor cells [32]. Rotondi et al. demonstrated that metformin inhibits the secretion of CXCL8 stimulated by TNF-α in primary cultures of human thyroid cells obtained from normal parenchyma and from papillary thyroid cancer, in a dose-dependent manner [33]. Figure 1 summarizes the mechanisms described before.

### 2.2. Proposed Indirect Antitumor Mechanisms Related to Metformin in DTC

It has been described that cancer cells are able to survive in an adverse tumor microenvironment with limited nutrients by reprograming canonical biochemical pathways to provide the necessary precursors of proteins, nucleic acids, and membrane lipids via shifts from anaerobic to aerobic glycolysis [16]. Bikas et al. explored molecular mechanisms related to the thyroid cancer cells response to glucose deprivation and metformin treatment. Their results showed that the glucose level in the growth media affects the efficacy of metformin as an antitumoral agent. In low-glucose media, metformin induces autophagy and oncosis, via AMPK-dependent and AMPK-independent mechanisms. Metformin induces glycolytic activity in thyroid cancer cells by increasing its dependence on glucose in the extracellular milieu and sensitizing them to a pharmacological inhibitor of glycolysis. In addition, metformin inhibits the expression of the master regulator of aerobic glycolysis (pyruvate kinase M2 gene (*PKM2*)), whose expression is elevated in tissue samples from thyroid cancer recurrences [34]. However, the molecular mechanisms involved in the metabolic shift that leads to higher glucose uptake by tumoral cells are poorly defined. Other associated mechanisms are the downregulation of type 1 glucose transporter (GLUT1) and hexokinase (HK2) expression, which reduce the rate of glucose metabolism in PTC in vitro and in vivo [35].

Another indirect mechanism proposed is that the use of metformin reduces TSH levels in patients with IR and elevated TSH. Some authors have suggested that the TSH-lowering effect of metformin is related with the activation of AMPK [36]. It has been hypothesized that metformin changes the affinity and/or quantity of thyroid hormone receptors, increases the central dopaminergic tone, and induces activation of the TSH receptor, enhancing the pituitary effects of thyroid hormones [37]. Indeed, it may be plausible that any central effects of metformin on the thyrotropin-releasing hormone (TRH)/TSH regulation involve the AMPK system. This has been observed in a case report of thyroid hormones resistance, where the use of metformin decreased TSH levels through modulation of the hypothalamic–pituitary–thyroid axis activity, as well as in peripheral tissues and thyroid gland [38,39,40,41]. Furthermore, it has been suggested that metformin reduces TSH levels by inhibition of leptin. Leptin is a hormone produced by adipocytes that regulates energy balance in the brain and is involved in the expression of TRH, leading to an increase in the secretion of TSH [42].

Iodine uptake is a crucial step for radioiodine therapy for thyroid cancer. In rat thyroid epithelial cells, the activation of AMPK reduced iodine uptake and the Sodium/Iodine Symporter (NIS) at protein and mRNA levels; meanwhile, the pharmacological blockage of AMPK signaling increased iodine uptake (data confirmed in an animal model). Similar results were observed in a follicular rat thyroid cell line, where AMPK modulation of NIS depended on the cAMP response element (CRE) present in the NIS promoter [28].

One of the concerns of those in vitro studies is the supraphysiologic doses of metformin required, which are 100–1000-fold higher than therapeutic levels reached in the sera of patients with diabetes [43,44]. Because the usual human plasma concentrations of metformin is 0.01–0.3 mM, the extrapolation to clinical practice could be limited [15]. However, some authors have shown that despite lower doses of metformin, tumor volume can be reduced [18,45]. Table 1 presents the principal molecular pathways proposed of metformin effects in Differentiated Thyroid Cancer.

## 3. Clinical Evidence of Metformin Use in Thyroid Cancer

Within the different types of thyroid cancer, there is wider clinical evidence on the use of metformin on DTC. Meanwhile, in MTC and ATC, there are no clinical trials yet. As explained above, preclinical trials suggest that metformin can reduce thyroid cancer risk and tumorigenesis. The majority of thyroid cancers are detected as a small mass and have a good prognosis. This has led to changes in the treatment strategy, adopting less aggressive approaches. We must consider that metformin is not as potent as a chemotherapeutic agent, but it has an important role in prevention, the reduction in progression, and as an adjuvant to chemotherapy treatment [46].

### 3.1. Thyroid Cancer Risk Reduction

Insulin resistance or hyperinsulinemia are risk factors for many kinds of cancers, such as melanoma, endometrial, hepatocellular, colorectal, breast, and lung cancer. Hyperinsulinemia increases cell glucose uptake and oxidative stress, and promotes thyroid cell growth, leading to hyperproliferation, carcinogenesis, and development of malignant tumors. It has been related to the thyroid nodules size in nodular disease, as well as tumoral size in patients with thyroid cancer [14,42].

In patients with hyperinsulinemia, an increased expression of tumor markers such as the proto-oncogenes: human epidermal growth factor receptor-2 (*HER-2*) and B-cell lymphoma-2 (*Bcl2*), were identified; meanwhile, tumor suppressor gene *p53* was significantly decreased [10,14].

Zhao et al. in a recent meta-analysis that included 14 articles with 2024 cases and 1460 controls studied the association between IR and thyroid carcinoma, and observed that patients with thyroid carcinoma had higher levels of the homeostatic model assessment for insulin resistance (HOMA-IR) than those without thyroid carcinoma, concluding that IR and hyperinsulinemia increase the risk of carcinoma with an OR of 3.16 (95% CI 2.09, 4.77) compared with those without IR using a fixed-effect model by a mild heterogeneity (chi^2^ = 5.23, *p* = 0.16, and I^2^ = 43%) [14]. That meta-analysis supports the relationship between IR and risk of thyroid cancer that was mentioned previously.

Meanwhile, some clinical observational trials identified metformin as a protective factor against DTC. Tseng et al., in Taiwanese patients with T2D, observed that ever-users of metformin had an adjusted HR of 0.683 (IC 0.598–0.78, *p* < 0.0001) for cancer development, in comparison with never-users. They reported that a decreased risk could be observed with a cumulative duration of 9 months of metformin use or a cumulative dose of 263,000 mg [42]. Similarly, in a retrospective cohort in Korean population, Cho et al. found that metformin use had an HR of 0.69 (CI 0.60–0.79, *p* < 0.001) for cancer, and that effect was stronger, reaching a higher cumulative dose (>529,000 mg) or a longer use (>1085 days) [46].

Table 2 shows a summary of the clinical studies related to the treatment of metformin in patients with DTC.

### 3.2. Metformin and TSH Suppression

One of the indirect proposed mechanisms for the reduction in the risk of thyroid cancer with the use of metformin observed in patients with IR and elevated TSH levels, is that metformin reduces TSH concentrations.

Mousavi et al. conducted a single-blind randomized controlled trial evaluating the effects of short-term treatment with metformin on TSH levels in patients without diabetes and DTC. They concluded that there is no benefit of its use at a dose of 500 mg daily for 3 months, in the reduction in TSH concentration [6].

However, in a meta-analysis by Lupoli et al., they observed that metformin induces a reduction in TSH levels both in overt (receiving thyroxine replacement) and subclinical hypothyroidism (without thyroxine replacement), with a mean decrease of 1.08 μUI/mL and 1.59 μUI/mL, respectively. In contrast, no change in TSH levels was found in euthyroid patients. Furthermore, the basal TSH value correlates directly with the degree of TSH change during metformin treatment (*r* = 0.955, *p* = 0.001) [41].

### 3.3. Role of Metformin as Co-Adjuvant Therapy in Radioactive Iodine Therapy in DTC

DTC has an excellent prognosis, with a 5-year survival rate of 98%. Despite low mortality, it requires an aggressive treatment in order to avoid the development of metastases. The most useful approach is a total thyroidectomy with further administration of high doses of radioactive iodine (RAI). RAI could be administered multiple times and doses, and, despite being mostly innocuous, could have several side-effects on other types of cells [49].

There is some evidence that metformin increased thyroid cancer cells sensitivity to radiation due cell redifferentiation, with a subsequent increase in RAI uptake. The mechanism involved is the attenuation of prosurvival signals, activation of AMPK, and inhibition of mTOR signaling, leading to the complete blocking of p70S6K/pS6 [47].

A side-effect of RAI treatment is hematopoietic toxicity, manifested as neutropenia, anemia, and low platelet count, as well as leukemia and bone marrow aplasia. Here, metformin has demonstrated radioprotective effects. Bikas et al. conducted a retrospective analysis comparing complete blood counts (CBC) values in DTC patients who were and were not taking metformin. They found that the white blood count (WBC) was higher in those using metformin (*p* < 0.0001), especially absolute neutrophil count (ANC). Furthermore, metformin users had a faster recovery in their blood counts. This effect was independent of metformin doses [50].

Currently, there is a clinical trial open for enrollment, focused on metformin as an add-on therapy to RAI (NCT03109847). It was designed to assess if metformin can mitigate the myelosuppressive effects of RAI and if it induces a faster recovery of white blood cell count to baseline values. The estimated date for study completion was January 2022, so its results may soon be published [49].

### 3.4. Role of Metformin in Thyroid Cancer Metastases

Despite multiple treatment options, local recurrences are found in 5–20% of patients with DTC, with two thirds being located in cervical lymph nodes. The recurrence of DTC is associated with various factors such as initial therapy, age, tumor size, gross soft tissue invasion, and distant metastases [48]. Approximately 50% of patients with DTC who present with distant metastases die within 5 years of initial diagnosis, despite treatment with surgery and RAI. Additionally, only 6.9 to 16% of patients with distant metastases reach complete remission. In addition to advancements in chemotherapy, metabolic interventions such as ketogenic diet, high-dose vitamin C, and metformin have been proposed as less toxic co-adjuvant therapies, targeting the reprogramming of glucose metabolism of advanced cancers [47,49].

A single-center observational study by Klubo-Gwiezdzinska et al. concluded that age, locoregional metastases, distant metastases, and the lack of treatment with metformin of patients with diabetes were associated with an increased risk for shortened progression-free survival. They observed that metformin-treated individuals had a smaller tumor size, and the lack of metformin treatment decreased the likelihood of complete response, suggesting its potential inhibition for tumor growth via the activation of AMPK and down-regulation of p70S6K/pS6 [47]. Another retrospective study conducted by Jang et al. in patients with diabetes with advanced DTC and lymph node metastases concluded that treatment with metformin is associated with a longer disease-free survival in comparison with those without metformin [48].

### 3.5. Role of Metformin in Thyroid Nodules

Thyroid nodules are common disorders, with a prevalence of 4–7% when diagnosed via palpation and 10–41% diagnosed via ultrasound [51]. Nowadays, its diagnoses have increased because of the use of newer and highly sensitive techniques. Some risk factors for the formation of thyroid nodules are iodine deficiency, female gender, age, tobacco use, genetic factors, and IR [52].

Some reports have identified a higher prevalence of thyroid nodules in patients with T2D [51]. IR increases visceral fat accumulation, which enhances TSH secretion mediated by leptin, and then TSH and IGF-1 act synergically to induce thyrocyte proliferation [52].

Anil et al. conducted a prospective interventional trial in euthyroid patients with IR. In this study, the use of metformin (1700 mg/day for 6 months) reduced thyroid volume (22.5 ± 11.2 mL vs. 20.3 ± 10.4 mL, *p* < 0.0001) and nodule size (12.9 ± 7.6 mm vs. 11.7 ± 7.2 mm, *p* < 0.0001), suggesting its antiproliferative effect [53]. Sui et al. conducted a systematic review and meta-analysis that included 7 studies with 240 patients evaluating the relationship between metformin therapy and reduction on thyroid nodules. The mean time of metformin use was 3–12 months, and the doses were 1500–2000 mg/day. They identified a significant reduction in nodule volume after metformin therapy (SMD −0.62, 95% CI −0.98 ∼ −0.27, *p* = 0.006), suggesting that this was related with an improvement in IR and a direct antiproliferative effect via suppression of the hypothalamic–pituitary–thyroid axis activity [51]. In a meta-analysis and systematic review conducted by He et al. that included 189 patients, metformin reduced thyroid nodule size, and TSH and HOMA-IR levels. However, the included studies had a low or very low quality of evidence and moderate or high heterogeneity. In that report, they concluded that metformin might have an important role as an adjuvant therapy to reduce the growth of benign and malignant thyroid neoplasms [10].

## 4. Conclusions

Metformin inhibits the growth and migration of thyroid cancer cells directly by mechanisms related to AMPK and mitochondrial respiration, and indirectly through effects on TSH levels and metabolic parameters associated with a less favorable environment for cell proliferation, as well as potentiating the effect of certain chemotherapeutic drugs. This suggests that metformin has an important supportive effect in patients with thyroid cancer and metabolic diseases such as insulin resistance or diabetes and may be an adjuvant treatment for differentiated or poorly differentiated thyroid cancer.

## Figures and Tables

**Figure 1 biomolecules-12-00574-f001:**
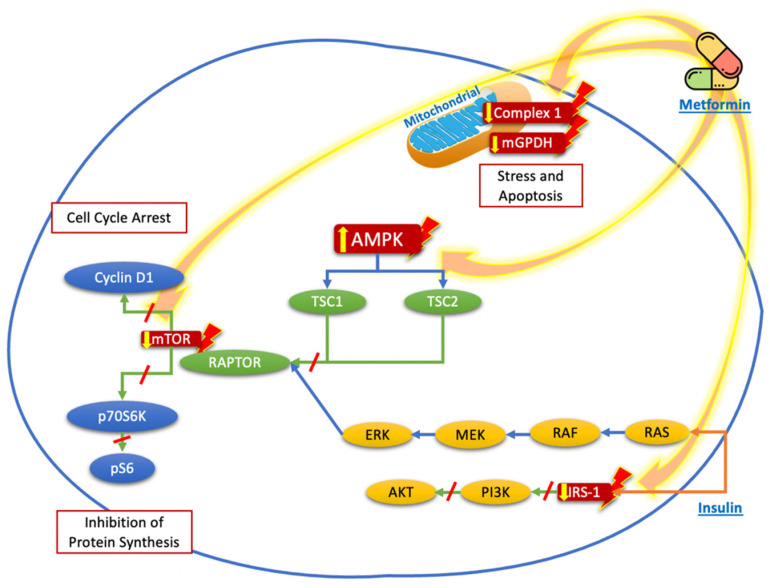
Proposed targets of the molecular mechanisms of metformin to reduce the proliferation and growth of thyroid cancer cells. Metformin acts mainly through nuclear stimulation of AMPK, which decreases the activation of TSC2 and the activation of mTOR, with the subsequent reduction in cyclin D1 and p70S6K/pS6 signaling, resulting in the inhibition of protein synthesis and cell cycle arrest. The blockade of IRS-1 phosphorylation decreases the signaling of the PI3K/AKT pathway, which also contributes to the reduction in mTOR pathway activation. Likewise, the direct effects of metformin on the mitochondria, by reducing Complex 1 or mGPDH, decreases energy production and cell stress and leads to cell apoptosis.

**Table 1 biomolecules-12-00574-t001:** Metformin pathways/targets identified by in vitro studies.

References	Pathways/Target	Effect
Mitochondrial action		
Thakur et al. [29]	Inhibition of mitochondrial complex 1 and mitochondrial glycerophosphate dehydrogenase (mGPDH)	Reduction in oxidative phosphorylation, decreased energy production, cell stress, and tumor cell apoptosis
Mechanisms involved in the inflammatory state		
Rotondi et al. [33] Bauerle [31]	Inhibits interleukin 8 (CXCL8) stimulated by tumor necrosis factor alpha (TNF-α) which decrease nuclear factor κB (NF-κB)	Reduced growth and progression of thyroid cancer
AMPK-dependent		
Chen et al. [17] Cho et al. [18] Hanly et al. [19]	Activation of AMPK triggers tuberous sclerosis complex 2 (TSC2), which inhibits the mTOR signaling pathway and hampers the activation of ribosomal protein S6 kinase beta-1 (p70S6K/pS6) and reduction in cyclin D1	Inhibition of protein synthesis and cell cycle arrest
Pierotti et al. [22]	AMPK activation phosphorylates an inhibitory serine residue in the insulin receptor substrate-1 (IRS-1), leading to downregulation of insulin-like growth factor 1 receptor (IGF-1R), which decreases the signaling of phosphatidylinositol 3-kinase/protein kinase pathway (PI3K/AKT), leading to the reduction in mTOR pathway activation	Inhibition of protein synthesis and cell cycle arrest
AMPK-independent		
Hadad et al. [23]	Tumor necrosis factor receptor 1 (TNFR1) and G1/S checkpoint regulation	Reduction in cancer cell growth
Kourelis et al. [24]	Inhibition of unfolded protein response (UPR)	Apoptosis, prevents angiogenesis, and induces toxicity on cancer stem cells

**Table 2 biomolecules-12-00574-t002:** Effects of metformin on thyroid cancer as reported in clinical studies.

Reference	Study Design	Objective	Patients Characteristics	Metformin Dose	Duration of Treatment or Follow Up	Conclusions
Tseng et al. [42]	Clinical observational trial	To investigate the association between metformin use and thyroid cancer risk.	795,321 metformin users and 619,402 non-metformin users, Taiwanese patients with T2D.	Cumulative dose of 263,000 mg.	9 months	Metformin decreased thyroid cancer risk by 32%.
Cho et al. [46]	Retrospective cohort study	To investigate the association between metformin and thyroid cancer development.	Korean population: 128,453 metformin users and 128,453 non-users.	Mean cumulative dose of 868,169 (±563,221) mg.	1633 (±915) days	Metformin reduced risk cancer by 31%.
Klubo-Gwiezdzinska et al. [47]	Single-center observational study	Whether the efficacy of conventional treatment of DTC is affected by therapy with metformin in patients with diabetes.	Patients with diabetes treated (n = 34) or not (n = 21) with metformin and control patients without diabetes (185).	500–2000 mg/day.	4.4 (±3) years	Age, locoregional metastases, distant metastases, and lack of treatment with metformin were associated with increased risk for shortened progression-free survival. Metformin-treated individuals had smaller tumor size and better remission rates.
Jang et al. [48]	Retrospective study	To evaluate the clinical outcome of patients with diabetes and DTC according to metformin treatment.	60 patients with diabetes and 201 control patients with DTC after total thyroidectomy.	Mean dose of 979 mg.	7.4 (±4.8) years	Metformin treatment was associated with longer disease-free survival.

T2D: Type 2 Diabetes Mellitus, IR: Insulin resistance, DTC: Differentiated Thyroid Cancer.

## Data Availability

Not applicable.

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
