# Peer review of "Metformin in Differentiated Thyroid Cancer: Molecular Pathways and Its Clinical Implications"

_biomolecules, 2022, doi:10.3390/biom12040574_

Round 1
Reviewer 1 Report
This manuscript titled “Metformin and Differentiated Thyroid Cancer: Molecular Pathways and its Clinical Implications” is interesting. The authors summarize the current clinical evidence of metformin use in thyroid cancer. Unfortunately, the main part of the paper describes the clinical aspects of metformin in thyroid cancer, but not, as the title suggest, the molecular pathways. The authors should rebuild the manuscript and show the major mechanisms involved in effects of metformin on thyroid cancer. In fact, a detailed understanding of molecular aspects of metformin and thyroid cancer is not presented, particulariy the following should be clearly mentioned and/or illustrated. Finally, only 41 references included in this review paper.
- Metforin and thyroid function including the effect of metformin on TSH suppression
- Insulin along with TSH functions as a growth factor and stimulates thyroid cell proliferation (TSH-insulin/IGF1) in the pathogenesis of thyroid morphological abnormalities (the action of IGF-1/insulin by PI3K signaling in thyroid)
- The antimitogenic activity in thyroid cancer
- Metformin – AMPK-mTOR pathways to inhibiter tumor growth
- Akt-mTOR activation is essential for the activation of innate immune cells and tumor-associated macrophages
- Metformin reduced p-ERK, and also increased expression of BIP, CHOP and caspase-12
- The effect of metformin in iodide uptake in the thyroid cancer context
Reviewer 2 Report
The authors review the molecular pathways targeted by metformin that may have clinical implications in the treatment of differentiated thyroid cancer.
Major comments:
As thyroid nodules are being diagnosed with increasing frequency with the use of newer and highly sensitive imaging techniques , it would be interesting to also elaborte in more details the evidence that metformin may have a role in the management of thyroid nodules.
Language changes and grammar corrections are required.
Line 39: I would suggest to include that metformin is not metabolized by cytochromes P450 in the liver so that drug transporters have the major role in its pharmacokinetics. Besides OCT1, I suggest to also mention the transporters involved in metformin excretion via kidneys.
Figure 1. Proposed checkpoints - would it not be better to say that these are the proposed targets? Namely, the therm checkpoint usually refers to the points in the cell cycle that control the progression to the next stage of cell division.
Table 1 summarizes nicely the effects of metformin on thyroid cancer as reported in clinical studies. It would be nice to have a similar table that would summarize the metformin targets / pathways identified by in vitro studies.
Minor comments:
Abstract and Line 75: popularly studied – please rephrase;
Abstract : Please tone down the statement: This suggests that it has an important supportive effect in patients with thyroid cancer >> it may have an important supportive effect
Please use the terms: radioactive iodine (RAI) or 31I treatment consistently.
Round 2
Reviewer 1 Report
The authors have addressed my comments. I have no further comments and recommend it for publication.
Author Response
Thanks.